# Evolution Behaviors and Reduction Mechanism of Curing Residual Stresses in GLARE Laminates under a Hot-Pressing Condition

**DOI:** 10.3390/polym14101982

**Published:** 2022-05-12

**Authors:** Huaguan Li, Hao Wang, Junxian Xiang, Zhaoxuan Li, Xi Chen, Jie Tao

**Affiliations:** 1Jiangsu Key Laboratory of Advanced Structural Materials and Application Technology, Nanjing Institute of Technology, Nanjing 211167, China; lizhaoxuan@njit.edu.cn; 2College of Material Science and Technology, Nanjing University of Aeronautics and Astronautics, Nanjing 211106, China; wanghao@nuaa.edu.cn (H.W.); xiangjunxian@nuaa.edu.cn (J.X.); nuaa_mst_cx@nuaa.edu.cn (X.C.)

**Keywords:** GLARE, residual stress, fiber Bragg grating sensor, numerical simulation

## Abstract

Nowadays, variable preparation, forming and processing methods of fiber metal laminates are constantly developing to meet the requirements of different application fields, hence the characteristics and evolution of residual stresses under different manufacturing conditions deserve more attention. In this work, the evolution behaviors of curing residual stresses in GLARE under a hot-pressing condition were studied, and the residual stress reduction mechanism was also explained. Results suggested the FE prediction models of the entire cure process, verified by the fiber Bragg grating (FBG) sensors, were more precise than the traditional elastic model. Moreover, the stress evolution during the cure process mainly occurred in the cooling stage, in which the different coefficient of thermal expansion (CTE) of aluminum and GFRP played a major role. Meanwhile, curing shrinkage stress in the GFRP layer during the holding stage at curing temperature obviously influenced the final stress level. The residual stresses in GFRP layers differed by 9.6 MPa under a hot-pressing and autoclave condition, in which the convection heat transfer condition played a major role as it caused lower thermal stress in the holding stage and a smaller temperature gradient in the cooling stage. Considering this, a lower cooling rate could be a feasible way to obtain GLARE with lower residual stress under a hot-pressing condition.

## 1. Introduction

GLARE laminates, manufactured by alternating aluminum alloy sheets and S-glass fiber prepregs [1], possess excellent fatigue and impact properties [2,3,4,5,6]. Nowadays, GLARE has been used in the fuselage and wings of aircrafts [7,8], and is also expected to be used in rail transit [9]. However, the cured GLARE laminates still suffer severe curing residual stresses, which can obviously influence the subsequent forming, machining and assembling processes [10,11,12,13]. Therefore, efficient reduction methods of curing residual stress are always necessary. In recent years, many preparations, forming and processing methods of GLARE have been constantly developed to meet the requirements of different fields: hot pressing, a non-autoclave preparation method, is now often used to manufacture FMLs products [14,15]. Additionally, shot peening, a kind of flexible forming process, is used to realize complex forming of FMLs. However, the characteristics of residual stresses are distinctly influenced by such stress-dominated processes [16,17], hence the research on the generation and evolution of curing residual stress becomes more significant. In further studies, the values of residual stress in GLARE should be precisely measured, and the monitoring of stress evolution should also be achieved.

Finite element analysis (FEA) is an ideal approach to recognize the composition and evolution behaviors of curing residual stresses during preparation and could also modify the preparation craft to control the stresses level [18,19]. For GLARE laminates, huge difference of CTE between aluminum alloy and GFRP prepregs lead to major residual stresses during the cooling process. However, the effects of cure phenomena and transformation of thermoset epoxy used as the matrix of GFRP prepregs were barely considered. Hence, besides the major part of residual stresses caused by temperature gradient, exact prediction of residual stresses should consider the evolution of the epoxy stiffness and the chemical shrinkage of the epoxy before the cooling process. Traditional FEA method for residual stress mainly based on the elastic model which merely considered the residual stress developed during the cooling stage and assumed no stress development prior to completion of the curing [20]. The elastic models combined with the classic lamination theory (CLT) could provide a good prediction of residual stress for thin laminates [21]. However, the elastic models neglected the evolution of residual stresses before the cooling stage and were hence unable to reflect the complexity of temperature and curing degree. Nowadays, FE simulation combining user subroutine achieved the stress evolution prediction of the entire cure process. Anxin et al. [22] established FE models considering heat transfer and resin behaviors to study the process-induced distortions of thermoset composites, in which the behaviors of resin and CFRP were defined by UEXPAN and UMAT. In this way, stress evolution models of the entire cure process could better reflect the actual behaviors compared to previous elastic models, and corresponding improvement methods could also be easily proposed.

Moreover, the simulated results should be verified by reliable experimental method. However, previous measuring methods such as layer removal and X-ray diffraction (XRD) [23] mainly focus on the final thermal residual stresses and are unable to describe the stress evolution in the entire cure process. In recent years, FBG sensors have been widely used to monitor the life cycle of composites due to immunity to electromagnetic interference, relative lightweight, durability, signal stability and suitability for wavelength multiplexing [24,25]. The data from FBG sensors can be considered as actual temperature and strain values inside the material, which are accurate and reliable to verify the validity of FE models. Qinglin et al. [26] successfully explored the stress and distortion evolution during thermocure processes of carbon fiber/epoxy composites combining numerical models and FBG sensors. For FML structures, FBG sensors can also be embedded between layers to provide in situ monitoring of stress evolution without significant effects on the global mechanical behavior of the laminate in view of the extremely small physical size of an optical fiber [27,28].

In this work, the evolution of curing residual stresses during the cure process of GLARE was simulated using sequential coupled FE models. Then, FBG sensors were embedded into the laminates to monitor the variation of the temperature and strain, and the simulation results were further verified. This way, the stress evolution behaviors in the cure process were analyzed in detail. Finally, the stress evolution behaviors under hot pressing and autoclave were compared, and a possible stress reduction method for hot-pressing craft was proposed. The analysis presented in this paper is a novel approach on GLARE laminates which can also be extended to any FML structures. In fact, the residual stress always exists and changes during the preparation, forming, machining, connection and service of FMLs. This study intended to accurately predict the formation process of residual stress in the preparation process through the establishment of the finite element model, which laid a foundation for the study of residual stress in the subsequent processing process of the laminate. Additionally, we have been trying to make it possible for FMLs to be applied in fields other than aviation, such as automobile and rail transit, by developing the low-cost preparation method. However, the residual stress caused by different preparation methods needed to be concerned.

## 2. Finite Element Models

The GLARE laminates were composed of alternating three layers of aluminum alloy sheets (2024-T3, 0.3 mm) and four layers of S2-Glass/FM94 prepreg (unidirectional, 0.125 mm), and the stacking sequence was [Al/0°/0°/Al/0°/0°/Al]. The parameters of each ply are shown in Table 1.

The FE models were established in ABAQUS based on the GLARE laminate with a dimension of 300 mm × 300 mm, the basic settings of the laminate are presented in Figure 1. In total, ¼ of the laminate was built to simplify the computation, and the XSYMM and YSYMM constrains were, respectively, added to the adjacent sides. Modelling was based on CLT incorporating the effects from the whole cure cycle including heating, curing and cool-down parts. Two sequential FE models were established to simulate the entire cure process. Firstly, a thermo-chemical model using heat transfer procedures was used to obtain the variation of temperature and curing degree during the cure process. Then, the properties’ evolution and cure shrinkage variation were accomplished using user subroutine UMAT based on the above thermo-chemical results. During the entire cure process, the overflow of resin and the thickness variation of the laminate were ignored, and the GFRP prepreg was in rubbery state with negligible viscoelastic relaxation as it was cured above its glass transition temperature. Moreover, the stiffness of the prepreg layers and the interface bonding between the aluminum alloy and GFRP were both related to the curing degree. Through this model, the residual stresses during the cure process of the GLARE laminate became obtainable and this model was also suitable for other FML structures.

### 2.1. Thermal-Chemical Model

The surface metal layers are in direct contact with the top and bottom plates of hot pressing, and thus the temperature on the metal surfaces is defined the same as the curing craft. Except for the effects of thermal properties on the temperature field in GLARE laminates, the heat release of the resin during the cure process should also be considered. Reaction heat of the resin generates with the increase in the curing degree, and the quantity of heat depends on the inherent properties of the resin. Hence, a thermo-chemical model is needed to characterize the evolution of temperature and cure degree throughout the laminate during the entire cure cycle. The process of heat transfer can be described by the Fourier’s heat conduction equation as:(1)$${\rho C}_{p}\frac{\partial T}{\partial t}=\frac{\partial }{\partial x}\left(k_{x}\frac{\partial T}{\partial x}\right)+\frac{\partial }{\partial y}\left(k_{y}\frac{\partial T}{\partial y}\right)+\frac{\partial }{\partial z}\left(k_{z}\frac{\partial T}{\partial z}\right)+Q$$
where *ρ*, *C_p_*, *k_x_*, *k_y_*, *k_z_* denote the density, specific heat and anisotropy thermal conductivities of the composite, respectively. *T* is the temperature at time *t* and *Q* is the internal heat generation due to the resin curing reaction.

Two kinds of curing kinetic models are usually used to present the relationship between the resin chemical reaction, temperature and cure degree: phenomenological model and mechanistic model. Phenomenological model has been widely used since it focused on the overall reaction and ignored the specific reaction of active substances in cure process. The parameters of the kinetics model for the curing of epoxy FM-94 have already been reported by the other authors [29]. Cure rates were calculated from differential scanning calorimetry (DSC) measurements and fitted to the Kamal reaction rate equation (Equation (2)) with parameters given in Table 2.
(2)$$\frac{\mathrm{dq}}{\mathrm{dt}}=k_{0}{\;e}^{{-E}_{A/}\mathrm{RT}}q^{m}{\left(1\;-\;q\right)}^{n}$$

In the above equation, *R* is the universal gas constant and *T* is the temperature in Kelvin. *E_A_* is the activation energy, *k*_0_ is a coefficient and *m* and *n* are two constants.

In the above FE model, heat transfer procedures were selected to simulate the thermal-chemical behavior of GLARE laminates, and the thermal boundary condition was set by subroutine DISP, meanwhile the heat release of the resin and the curing degree at any time *t* were together subscribed by user subroutine HETVAL and USDFLD. Initial cure degree was set as 0.0001 to meet the demand of the program, and the initial temperature of the entire laminate was 20 °C. In total, this model contained 4400 DC3D8 elements and 6174 nodes.

### 2.2. Stress Evolution Model

FM 94 epoxy, as a typical thermosetting resin, underwent a transition from a liquid to solid state during the cure process. In this process, the elastic modulus *E_m_* and shear modulus *G_m_* were regarded as
(3)$$\left\{\begin{array}{c} E_{m}=E_{m}^{0}\;,\;\;\;\;c<c_{gel}\\ E_{m}=\left(1-c_{mod}\right)E_{m}^{0}+c_{\mathrm{mod}}E_{m}^{\infty }\;,\;\;\;c\geq c_{gel}\\ G_{m}=\frac{E_{m}}{2\left(1+\nu_{m}\right)}\\ c_{mod}=\frac{c-c_{gel}}{1-c_{gel}} \end{array}\right.$$
where $E_{m}^{0}$ and $E_{m}^{\infty }$ represent the elastic modulus of the resin before and after curing, and $c_{gel}$ denotes the curing degree at the gel point of the resin.

During the entire cure process, the strain of the prepreg layer is composed of thermal strain *ε^th^* induced by temperature change, shrinkage strain *ε^sh^* induced by the shrinkage of the matrix resin and *ε^e^* induced by mechanical load. User subroutine UMAT defines the stress and Jacobi matrix ∂Δσ/∂Δε of the GFRP layer at the integral point in each increment step, and further calculates the *ε^e^* and stress evolution of prepreg layers. In this process, glass fibers are assumed to be evenly distributed in the prepreg, and the interface between the fiber and matrix is ideal and the void in the resin is ignored. Thus, the micromechanics method can be used to predict the macroscopic properties of the composite based on the properties and structural parameters of glass fiber and epoxy.

Previous FE models of residual stresses in FMLs mainly focused on the cooling process, hence the final results were only relevant to the temperature gradient and material properties of each ply. If the entire cure process is observed, the curing shrinkage strain should also be predicted based on the properties and distributions of GFRP prepregs.

Similar to an isotropic material, the shrinkage strain of pure resin is equal in all three principal directions. Then, the shrinkage of the resin at an increment step $\Delta \varepsilon_{r}^{\mathrm{sh}}$ could be expressed as:(4)$$\Delta \varepsilon_{r}^{\mathrm{sh}}=\sqrt[{3}]{1+\Delta V_{\mathrm{sh}}}-1$$

Similarly, the shrinkage of unidirectional GFRP prepreg at an increment step can be shown as:(5)$$\Delta \varepsilon_{1}^{\mathrm{sh}}=\frac{\Delta \varepsilon_{r}^{\mathrm{sh}}E_{r}\left({1-V}_{f}\right)}{E_{1f}V_{f}{+E}_{r}\left({1-V}_{f}\right)}$$
(6)$$\Delta \varepsilon_{2}^{sh}=\Delta \varepsilon_{3}^{sh}=\left(\Delta \varepsilon_{r}^{sh}+v_{r}\Delta \varepsilon_{r}^{sh}\right)\left(1-V_{f}\right)-v_{r}\left(1-V_{f}\right)\Delta \varepsilon_{r}^{sh}$$
where subscripts 1, 2 and 3 denote the principal directions of GFRP prepreg, subscripts *r* and *f*, respectively, represent resin and fiber, *E* is Young’s modulus, *ν* is Poisson’s ratio and *V_f_* is the volume fraction of fiber.

The CTE of the GFRP layer could be calculated based on the properties of glass fiber and epoxy, then *ε^th^* could be obtained combining the temperature gradient. This process mainly accomplished by subroutine UEXPAN, in which the thermal expansion at the integrate point was calculated.

In the stress evolution model, static-general procedures were selected to simulate the mechanical behavior of GLARE laminates, and the UMAT calculates the mechanical strain except the thermal strain and shrinkage strain caused by the temperature and curing degree. The non-mechanical strain caused by the temperature and curing degree in each incremental step is calculated by the subroutine UEXPAN. Moreover, UEXPAN also transmits variables such as temperature, cure degree and non-mechanical strain to UMAT. In total, this model contained 4400 C3D8 elements and 6174 nodes.

## 3. Experimental

The GLARE laminates were fabricated with layers of aluminum alloy sheets (2024-T3, 0.3 mm thickness per sheet) and S4-glass/epoxy prepregs (0.125 mm thickness per layer) by hand layup technique. Laminates were prepared by hot pressing, in which the temperature cycle was applied to the top and bottom surfaces of the laminate. All aluminum alloy sheets were treated before the lay-up process, including degreasing with acetone, alkali cleaning, acid cleaning and anodizing in a phosphoric acid electrolyte [29]. After the lay-up process, these stacks were cured under controlled temperature and pressure conditions. Figure 2 shows the hot-pressing curve for curing the laminates.

The FBG temperature and strain sensors (the basic parameters are listed in Table 3) were embedded longitudinally between the GFRP layers during the lay-up process, and were 10 mm and 150 mm distance from the edge of the laminate, respectively. The schematic diagrams of GLARE and locations of embedded FBG sensors are presented in Figure 3a. The FBG temperature sensors were encapsulated in a steel capillary tube and their sensitivity constants in the range of 30–140 °C were estimated in an oven. Meanwhile, the strain sensitivity of FBG strain sensors were calibrated by the uniaxial tensile tests according to ASTM-D3090 standards. The central Point A in Figure 3a was regarded as an actual monitored point to verify the temperature and stress values in the FE models at the same location. Teflon tubes were used to protect the optical fiber from breaking in the egress location during the fabrication of the laminate, and the end of the optical fiber was connected with the interrogator shown in Figure 3b. At the same time, the edges of the stacked laminate were covered by peel ply tape and surrounded by a rectangular steel frame to reduce the disturbance to the sensor caused by resin overflow. After the cure process, only a little amount of resin spilled from the corner and the Teflon tube location (shown in Figure 3c), which meant that the flow of the resin inside the laminate was effectively limited.

The data obtained from FBG sensors represent the variation of the wavelength of reflected light ($\lambda_{B}$), which could be inferred combining the effective refractive index (*n_eff_*) of the fiber core and the Bragg period (Λ) of the grating:(7)$$\lambda_{B}{=2\;n}_{\mathrm{eff}}\Lambda \;$$

When the external temperature and pressure load change, the change in the reflected Bragg wavelength due to the axial strain *ε* and temperature change Δ*T* is expressed as:(8)$$\Delta \lambda_{B}=\lambda_{B\;}\left[\left(1-P_{e}\right)\varepsilon +\left(\alpha_{f}+\zeta \right)\Delta T\right]=K_{\varepsilon \;}\varepsilon +K_{T\;}\Delta T$$
where *P_e_* is the photoelastic coefficient, *α_f_* is the CTE of glass fiber and *ζ* is the thermo-optic coefficient. *K_ε_* and *K_T_* denote the strain and temperature sensitivity constants, respectively. For the FBG temperature sensor, the FBG is encapsulated in a steel capillary tube and is in strain-free condition [30]; therefore, Equation (9) can be simplified in terms of the temperature change as:(9)$$\Delta \lambda_{B}=\lambda_{B\;}\left(\alpha_{f\;}+\zeta \right)\Delta T=K_{T\;}\Delta T$$

Furthermore, the influence of thermal expansion on FBG strain sensors could be eliminated since the FBG strain and temperature sensors are embedded in the laminate synchronously. In this condition, the strain sensors could show the effects of chemical shrinkage and thermal expansion of the resin.

## 4. Results and Discussion

### 4.1. Evolution of Temperature and Cure Degree during the Cure Process

The relationships of temperature and curing degree with time in each layer are plotted in Figure 4. The temperature of the aluminum surface is the same as the preset temperature curve, but the temperature evolution of the internal layers exhibits obvious differences due to the variable heat transfer characters of each layer. Obvious delay in the heating and cooling processes occurs in the internal layers, which causes a temperature gradient through the thickness of the laminate. Moreover, the highest temperature value is beyond the preset 120 °C before the cooling process. The cure of epoxy has no development before 6000 s, and the highest curing rate occurs during 6000–9000 s. Meanwhile, delays in the curing degree in GFRP layers are also observed during this period since the cure rate is a function of temperature. Finally, the curing degree of each layer reaches the maximum after enough holding time at curing temperature.

The temperature gradient produced by the thickness of GLARE is a result of the different thermal conductivities of the GFRP and aluminum layers, which may lead to bigger residual stresses during the cooling process. Hence, the temperature differences through the thickness of GLARE were further studied in detail. As shown in Figure 5a, the temperature distribution nephogram at the timing *t* = 12,000 s was observed, and Path 1 was drawn along the thickness of the laminate at Point A, meanwhile the temperature gradient along Path 1 was shown in Figure 5b. Temperature is distributed symmetrically along the thickness and the gradient is about 2.8 °C (120 °C to 122.8 °C), which means the GFRP layer has a higher temperature when the laminate begins to cool down. This difference corresponds to the temperature delay in the GFRP layer at the same time and larger residual stresses may be produced. Therefore, enough holding time at the cure temperature is significant and can decrease the final residual stresses in FMLs.

The evolution of temperature at Point A monitored by the FBG temperature sensor is drawn in Figure 6. The temperature curve obtained by the FBG sensors is basically consistent with the preset temperature, but obvious undulation appears at the beginning of every dwell: the actual temperature value reaches 91 °C at the preset 80 °C dwell and 135 °C at the preset 120 °C dwell. At the 120 °C dwell, slight temperature fluctuations caused by the discontinuous heating characteristic of the equipment were also observed. It can be seen that the FBG temperature sensor is a much more accurate and convenient approach to obtain the actual temperature evolution and even the undulations in the laminate than the traditional temperature testing devices. Therefore, the monitored temperature can be considered as the actual temperature of the GFRP layer.

### 4.2. Stresses Evolution of GLARE during the Cure Process

The transverse stresses’ evolutions at Point A using the FE simulation and FBG sensor monitoring are shown in Figure 7, in which both simulated and monitored results show consistent tendency. At the beginning of the cure process, epoxy under high temperature is in viscous flow state. Stresses in the GFRP layer exhibited by the simulation and experiment keep small before the temperature reaches 120 °C, which means the increase in temperature during this period hardly affects stress value. Then, the expansion of epoxy becomes more obvious with temperature reaching to 120 °C. Meanwhile, the resin begins to cure rapidly and the curing shrinkage strain is also produced. Curing shrinkage strain is dominant during this period compared with thermal strain since the strain of the GFRP layer gradually evolves to be negative. The simulation and experimental results are more consistent at the 120 °C dwell, while there are frequent fluctuations on the FBG monitoring curves at the insulation process. During the cooling stage, the resin is completely cured and firm interface formed between FBG sensor and resin matrix, the stress increases rapidly with the decrease in temperature, which is mainly caused by the difference in the thermal expansion coefficient between the aluminum alloy and GFRP plies.

However, stresses obtained by simulation and FBG monitoring show some differences at the holding period at 40 °C and 80 °C. The stress value monitored by FBG sensor in this period reaches about 9 MPa, while the simulated data are below 5 MPa. This difference could be explained by the state of epoxy combining the curing degree curve. Although there is no firm bond between the sensor and the resin matrix, the strain monitoring using the FBG sensor is more easily affected by many factors because of its high sensitivity. On the one hand, the embedded FBG sensors bend slightly under the impact of flowing resin; on the other hand, as the temperature increases, the expanded resin also exerts a force on the FBG sensor. Hence, the FBG sensors exhibit higher values and frequent variation on stress.

The simulated result of stress evolution at the center of the surface aluminum alloy is shown in Figure 7b. During the heating stage, stress in the aluminum alloy layer is at a low level as the interface between the aluminum alloy and GFRP layer is weak. Then, firm combination forms between the layers with the increase in the curing degree in GFRP, and the behavior of the GFRP layer becomes more influential to the adjacent layer. Hence, tensile stress gradually increases during the cooling stage due to the larger CTE of aluminum. Although the stress evolution process in aluminum cannot be monitored, the final residual stress values can be measured by the layer removal method, and the experimental data are collected in Table 4. The average residual stress obtained by the layer removal method is 56.65 MPa, and corresponding stress in the GFRP layer is −101.97 MPa, which can verify the accuracy of FEA results.

The simulated final residual stress values and distributions are displayed in Figure 8. The residual stresses in the aluminum alloy layer and GFRP layer were 51.8 MPa and −92.4 MPa, respectively, and the stresses were uniform except for the edges. Residual stresses in the GFRP layer were compressive due to the lower CTE compared to the aluminum alloy. For comparison, the conventional residual stress simulating model based on elastic theory was also established, which only considered the cooling-down process, and the simulated residual stresses in the aluminum alloy and GFRP were 45.6 MPa and −97.1 MPa. On the basis of FBG monitoring data (−81.6 MPa), the accuracy of the FE models considering the entire cure process was improved by 5% compared to the conventional elastic FE models, and hence it is more reliable to modify present preparation craft.

### 4.3. Reduction Mechanism of Residual Stresses under Hot-Pressing Condition

Recently, hot pressing has been used more often than autoclaves in the preparation of FMLs due to lower costs, although autoclaves provide excellent loading mode and vacuum environment. In previous research, GLARE laminates were also prepared in autoclave and corresponding residual stresses were measured by the layer removal method, and the stress values (about −70 MPa in GFRP layer) were lower than the above results. The stress evolution FE models used earlier in this article were credible to display the composition of the final residual stresses, hence the residual stress reduction methods could be found by comparing the stress composition under hot-pressing and autoclave conditions.

The FE model of the cure process in autoclaves was established to compare with the results of the hot-pressing method. According to related literature, major differences between these two methods in the preparation of GLARE included the loading mode and thermal boundary condition, and the specific settings in the FE models were shown in Table 5.

The stresses’ evolution during the cure process in hot pressing and autoclave were firstly simulated, then a variable-controlling approach was used to explore the most influential factor on the evolution of residual stress. Each stress evolution curve is shown in Figure 9, and the final residual stresses of different models are listed in Table 6. The final residual stresses in the GLARE laminate cured in an autoclave exhibit lower residual stresses, although the laminate experienced a longer curing period. Compared to the stress evolution during hot pressing, additional load added on the sides of the laminate did not affect the final residual stress, but the change in thermal boundary condition had a great impact. It can be found that the stress changes during the cooling process were almost the same for the laminate cured in hot pressing and autoclave, hence the major difference appears in the holding stage of 120 °C, where the stress in hot pressing was larger than that in the autoclave condition. Hence, the difference in the final residual stress depends on the stress difference in the holding stage of curing temperature.

This phenomenon can be explained by the temperature evolution behavior at Point A, as shown in Figure 10. In the holding stage of 120 °C during the cure process, the shown stress values were composed of curing shrinkage stress and thermal stress of the GFRP layer, and the curing shrinkage stress dominated. In view of the excellent thermal conductivity of aluminum alloy, heat was easily transferred from the surface to the GFRP layer. Hence, the temperature at this moment depended on the thermal boundary condition applied to the surface of the laminate. In this respect, the heat transfer rate of convection heat transfer (film condition in FE model) was lower than the first boundary condition, which also meant the heat release of the resin was hard to spread out. Hence, the heating process of the GFRP layer under film condition delayed about 780 s and a higher temperature value also appeared in Figure 10, which affected the shrinkage and thermal expansion of the resin. In order to study the specific value of temperature and cure degree, the formation periods of curing shrinkage stress are marked in Figure 10.

As shown in Figure 10, the curing shrinkage stress appears at the heating stage, when the cure degree of the resin reaches about 0.3. The variation in temperature for the hot-pressing condition is 13.25 °C (from 108.45 °C to 121.70 °C) during this process, and for the film condition it is 16.12 °C (107.3 °C to 123.42 °C). The curing shrinkage of the resin is almost the same because the uniform change in cure degree, hence the GFRP layer under film condition has a higher thermal expansion strain due to a bigger temperature difference. Therefore, the GFRP layer under the film condition had a smaller stress value at the holding stage. During the cooling process, curing shrinkage stress together with thermal stress produced with the decrease in temperature composed the final residual stress. Under the circumstance of the same temperature drop, the GLARE laminate under the convection condition hence obtained lower residual stresses.

On the other hand, the temperature gradient in the thickness of the GLARE laminate during the curing process also affected the value of the final residual stresses. Figure 11 shows the temperature gradient along the thickness at Point A under a hot-pressing and autoclave condition. The temperature gap of the laminate in hot pressing is 17 °C (from 57 °C to 74 °C) at 18,000 s, while it is 8 °C (from 83 °C to 91 °C) under an autoclave condition. In this case, the difference in the thermal shrinkage between the aluminum alloy and GFRP layer was more obvious under a hot-pressing condition, and bigger residual stresses were produced. In this aspect, better residual stress characteristics may be achieved by a lower cooling rate for hot pressing.

Although the convection heat transfer condition obviously influenced the final residual stress value, the change in the thermal boundary condition was difficult for hot pressing. Hence, a lower cooling rate was applied to the stress evolution FE model to investigate its effect on the final residual stress value. As shown in Figure 12, residual stress in the GFRP layer decreased from −92.4 MPa to −88.8 MPa owing to the change in the cooling rate (from 0.5 °C/min to 0.3 °C/min). Meanwhile, the temperature gradient (8 °C) shown in the appended figure was lower than 17 °C in Figure 11a, which meant that the lower residual stress was caused by a lower temperature gradient. However, the effect of the cooling rate on residual stress was more limited compared to the effect of the thermal boundary condition, as it could not involve the formation of curing shrinkage stress. Moreover, the increased costs from the prolonged cooling period should be further considered.

On the basis of these conclusions, we can obtain the GLARE laminate with lower residual stresses using simple hot-pressing equipment by adopting a control program of lower cooling rate and enough holding period at the curing temperature. However, if the quality of GLARE cured in a hot pressing condition is expected to be exactly the same as that in the autoclave, measures must be taken to restrict the overflow of the resin and stacks should be vacuumized to reduce internal voids. Furthermore, the simulation and experimental methods in this work are also suitable for all kinds of fiber metal laminate structures.

## 5. Conclusions

(1)Sequential coupled FE models including the thermal-chemical model and stress evolution model were established to analyze the residual stresses of GLARE during the entire cure process, then the simulated results were successfully verified by embedded FBG sensors between GFRP layers. This way, the stress evolution behaviors in the cure process of GLARE were analyzed in detail.(2)The evolution of stress during the cure process mainly occurred in the cooling stage, in which the different CTE of aluminum and GFRP played major role. Meanwhile, curing shrinkage stress in the GFRP layer during the holding stage at curing temperature obviously influence the final stress level. Simulated residual stresses (−92.4 MPa) of GFRP were reliable compared to the monitored data (−81.8 MPa) from the FBG sensors. The prediction accuracy of residual stresses was improved by 5% compared to the conventional linear elastic prediction models (−97.1 MPa) which only covered the cooling process.(3)Between the hot-pressing and autoclave methods, the convection heat transfer condition was the key factor to obtain lower residual stresses as it caused lower thermal stress in the holding stage and a smaller temperature gradient in the cooling stage. However, additional loads on the sides of the laminate hardly affected the stress level in the cured laminate. Considering this, a lower cooling rate could be a feasible way to obtain GLARE with lower residual stress under a hot-pressing condition.

## Figures and Tables

**Figure 1 polymers-14-01982-f001:**
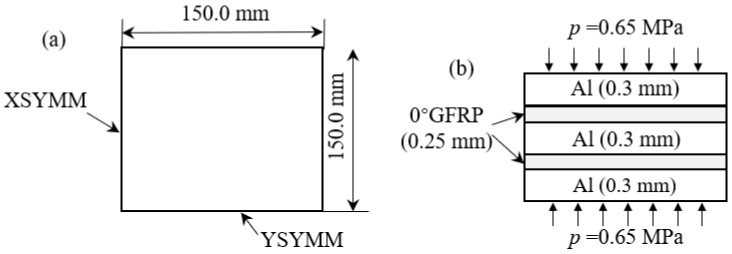
(**a**) Dimension and constraints of GLARE laminate; (**b**) Stacking sequence and load on laminate.

**Figure 2 polymers-14-01982-f002:**
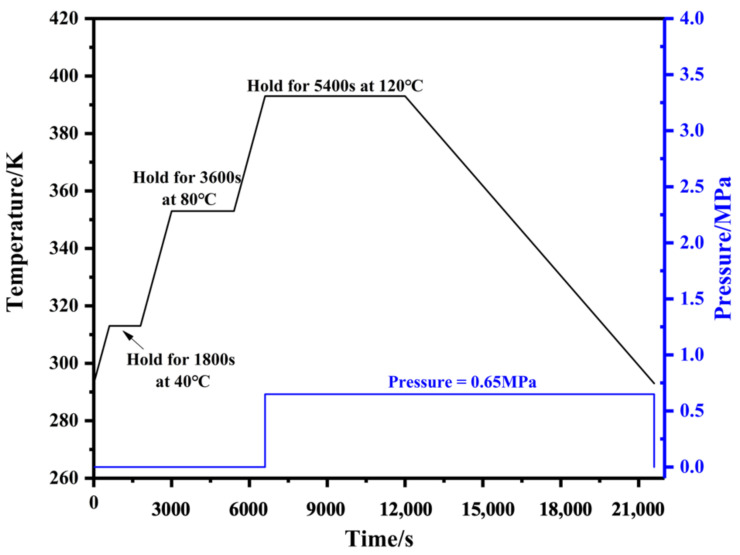
The temperature and pressure curves of cure process.

**Figure 3 polymers-14-01982-f003:**
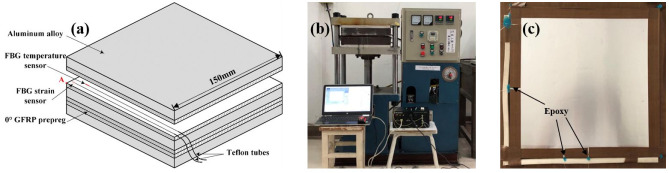
(**a**) The location of FBG sensors in 1/4 GLARE laminate; (**b**) Hot pressing and interrogator; (**c**) The cured laminate.

**Figure 4 polymers-14-01982-f004:**
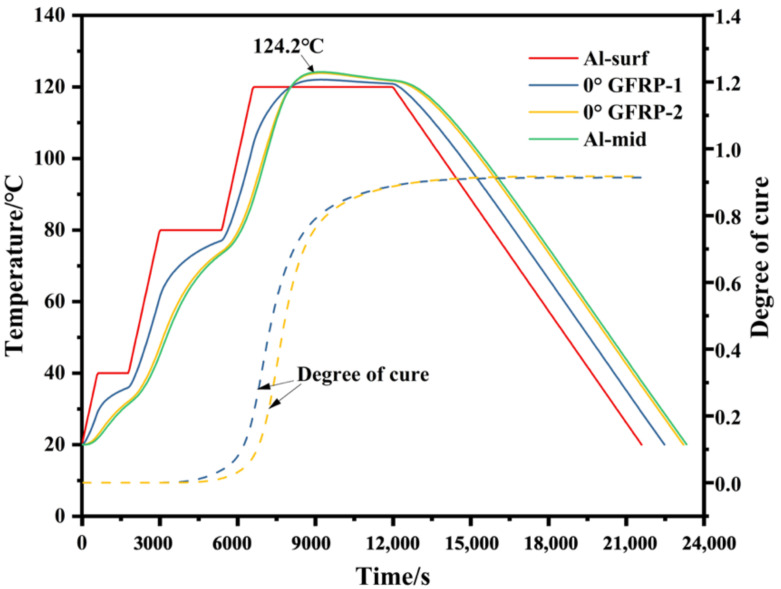
Simulated temperature and curing degree curves during the cure process.

**Figure 5 polymers-14-01982-f005:**
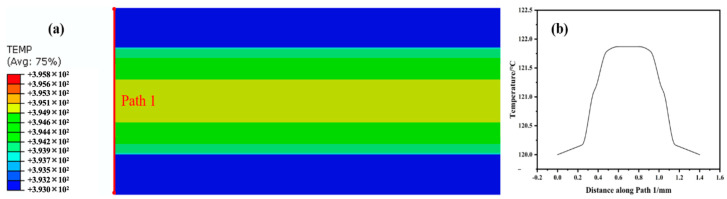
(**a**) The nephogram of temperature distribution along the thickness at *t* = 12,000 s. (**b**) The variation in temperature along Path 1.

**Figure 6 polymers-14-01982-f006:**
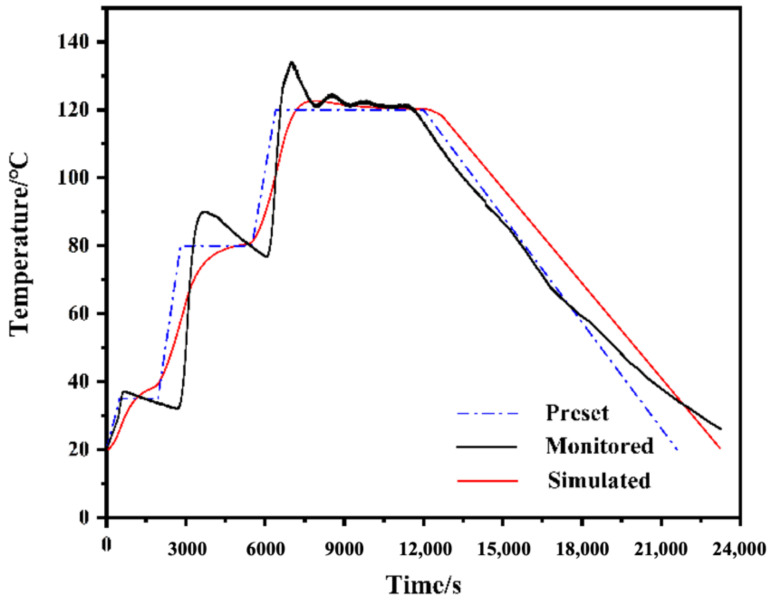
Comparison between preset, monitored and simulated temperature evolution.

**Figure 7 polymers-14-01982-f007:**
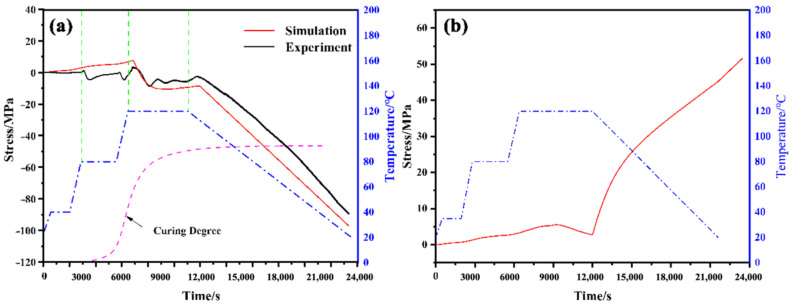
(**a**) The evolution of stress at Point A obtained by simulation and experiment; (**b**) The evolution of stress at the center of the surface aluminum alloy.

**Figure 8 polymers-14-01982-f008:**

The distribution of final residual stresses in (**a**) aluminum alloy layer and (**b**) GFRP layer.

**Figure 9 polymers-14-01982-f009:**
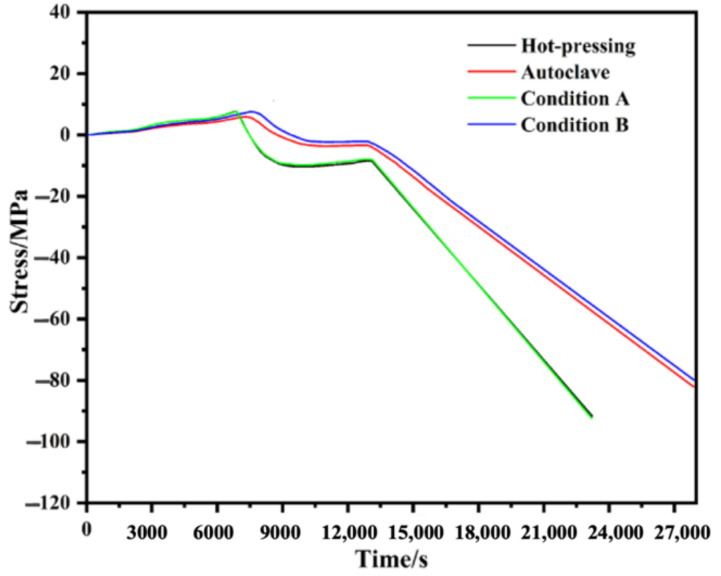
Influences of different process conditions on the evolution of residual stresses. Condition A: the external load was added on all the surfaces of the laminate; Condition B: the thermal boundary condition was replaced by convection heat transfer.

**Figure 10 polymers-14-01982-f010:**
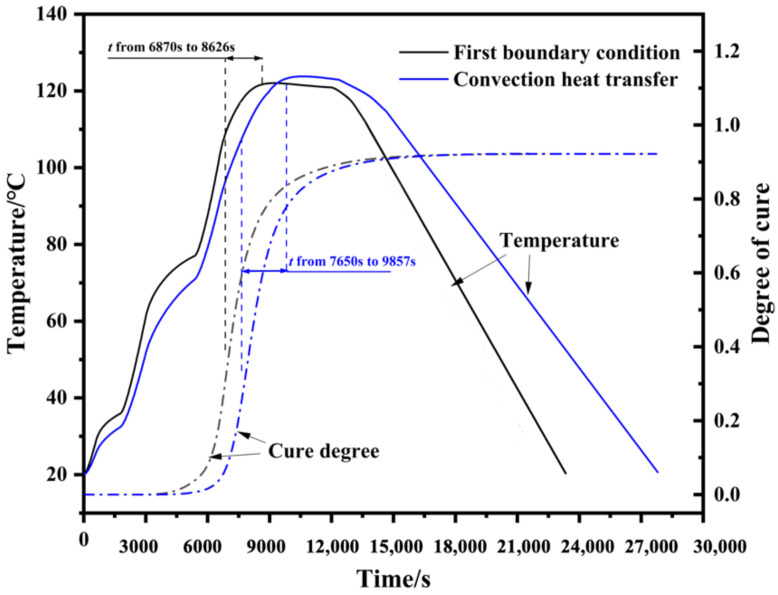
The evolution of temperature and curing degree in the GFRP layer under first boundary condition and convection heat transfer condition.

**Figure 11 polymers-14-01982-f011:**

Temperature gradient in the thickness of the laminate at Point A under (**a**) hot pressing and (**b**) autoclave at 18,000 s.

**Figure 12 polymers-14-01982-f012:**
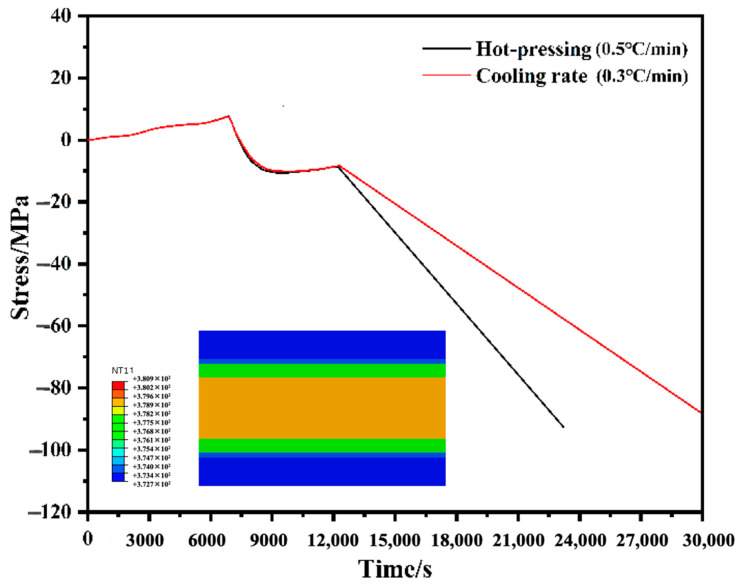
Comparison of stress evolution in the GFRP layer at the cooling rate of 0.5, 0.3 °C/min, and the appended figure displays the temperature gradient in the thickness of GLARE under the cooling rate of 0.3 °C/min at the timing *t* = 18,000 s.

**Table 1 polymers-14-01982-t001:** Material parameters for GLARE constituents.

Aluminum 2024-T3	E [MPa]	G [MPa]	ν	α [1/°C]	Conductivity [W/m/K]	Specific Heat [J/kg/K]
	72,400	27,600	0.33	2.3 × 10^−5^	120	875
S2-Glass/FM94 prepreg	E_1_ [MPa]	E_2_ = E_3_ [MPa]	G_12_ = G_13_ [MPa]	ν_12_ = ν_13_	α_1_ [1/°C]	α_2_ [1/°C]
	55,000	9500	5500	0.33	6.15 × 10^−6^	2.6 × 10^−5^
	Conductivity [W/m/K]			Specific Heat [J/kg/K]		
	0.75			834		

**Table 2 polymers-14-01982-t002:** Cure kinetic parameters for epoxy FM-94.

k_0_ [1/s]	E_A_ [J/mol]	m	n	H [J/g]
3.52 × 10^6^	6.75 × 10^4^	0.558	2.508	134.9

**Table 3 polymers-14-01982-t003:** Basic parameters of FBG sensors.

Parameters	Values
Central wavelength [nm]	1530–1560
Grating aera length [mm]	10
Reflectivity	91.87%
Diameter [μm]	150
Temperature sensitivity constant pm [°C]	10.61
Strain sensitivity constant pm [με]	1.22
Diameter of steel capillary tube [mm]	0.6

**Table 4 polymers-14-01982-t004:** The residual stress values obtained by the layer removal method.

Sample Number	Curvature Radius *ρ* (mm)	$\sigma_{\mathrm{Al}}\;(\mathrm{MPa})$
Sample 1	609.68	57.01
Sample 2	607.34	57.23
Sample 3	620.01	56.06
Sample 4	615.95	56.43
average	613.25	56.65

**Table 5 polymers-14-01982-t005:** Different settings of the cure process in autoclave and hot pressing in FE models.

Equipment	Loading Mode	Thermal Boundary Condition
Autoclave	Load was added on each surface of the laminate by uniform pressure.	Film condition is applied on all the surfaces. Film coefficient is 70 W/m^2^·K.
Hot pressing	Load was added on top and bottom surfaces by uniform pressure.	Temperature curve is directly added on top and bottom surfaces. The sides are under adiabatic condition.

**Table 6 polymers-14-01982-t006:** Final residual stresses in the GFRP layer under different conditions.

Different Conditions	Hot Pressing	Autoclave	Condition A	Condition B
Residual stress/MPa	−92.4	−82.8	−92.7	−82.2

## Data Availability

The raw/processed data required to reproduce these findings cannot be shared at this time as the data also form part of an ongoing study.

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
