# Peer review of "Evolution Behaviors and Reduction Mechanism of Curing Residual Stresses in GLARE Laminates under a Hot-Pressing Condition"

_polymers, 2022, doi:10.3390/polym14101982_

Round 1

Reviewer 1 Report

The work «Evolution Behaviors and Reduction Mechanism of Curing Residual Stresses in GLARE Laminates Under Hot-pressing Condition» is devoted very interesting topic. Modeling of different processes and properties is actual for current research. Need for mathematical modeling of GLARE laminates is good described in introduction. All models used are shown in equation form. Figures have good quality and presentability. The conclusions are well structured, which is convenient for readers. But I have one comment, namely

- The article is replete with formulas, but not everyone understands how they were used in the study of samples, that is, I recommend leaving only those formulas that were used to calculate.

Reviewer 2 Report

The authors reported results on the fiber metal laminates. They have studied the evolution behaviour of curing residual stresses in GLARE under  hot-pressing condition. The manuscript is well organized and supported by data in the framework of their expected outcome. However, by consulting carefully the scope of the Polymers, the topic of the present paper does not fit fully. Even though the materials the authors are reporting are considered somehow composites, still there was observed a very big difference between the topic proposed and the other already published.

Additionally, please find some comments/suggestions that may help improving the quality of the manuscript:

  1. The aim of the manuscript was not fully understood, and it needs more highlighting.
  2. Page 6 line 203 phrase “ Error! Reference source not” appeared as there is an error. The authors are advised to double check this phrase.
  3. The source of GLARE laminates was not clear enough; the authors are requested to provide details
  4. As the topic of the paper does not fit the Journal's scope, the authors are requested to argue why this paper should be published within.

Round 2

Reviewer 2 Report

The authors answered to the addressed queries and the manuscript was updated accordingly. Thank you.